# ON THE DECREASE OF GRANDMOTHER-ENTROPY IN AI

## ABSTRACT

Machine intelligence is best achieved today through deep learning. We propose a method to quantify the information content of activation values in a neural network layer for a given input. First, we map the activation of each node to a probability that reflects the likelihood that this node is selected among all nodes in the layer, where the more activation, the higher the probability. Then, we measure the information of the layer across channels or features, using conditional entropy, and we refer to it as grandmother-entropy. Empirical evaluation of ResNet-50 shows that the grandmother-entropy decreases as the input image propagates forward through the network layers. Moreover, the grandmother-entropy of the last convolution layer provides a reliable confidence measure to the classification result.

## 1 INTRODUCTION

### 1.1 INTELLIGENCE AND DEEP LEARNING

Deep learning LeCun et al. (2015) is today a central tool for building intelligent systems. By deep learning, we mean deep hierarchical neural networks trained with gradient descent. A common technique within this framework — skip connections — have become standard in most state-of-the-art models.

In this work, we explore the role of information in deep learning, paying particular attention to this technique. Our study is both theoretical and empirical. On the theoretical side, we introduce a measure of information for deep learning layers. On the empirical side, we evaluated this measure on ResNet-50 He et al. (2016), a widely used architecture that remains a benchmark in large-scale image classification. Rather than proposing a new architecture or application, our goal is to provide a clearer understanding of deep neural networks operations and to shed light on how classification models convey confidence in their predictions.

The study of information content in neural networks (NN) covers various aspects of data processing.

One direction of research treats the variables updated during learning, namely the weights and biases, as random variables and associates them with a measure of information, a tutorial Hoefler et al. (2021) provides the references.

Another direction considers the input vector $x$ as a random variable originating from a distribution $P(x)$ of all possible inputs for which the machine is learning. For example, images belonging to a given class can be viewed as originating from a specific class distribution. Extending this idea, the activation distribution in each layer, given an input $x$, is also modeled as a sample of a distribution on all possible activations of that layer. Representative studies include Gabrié et al. (2018); Saxe et al. (2018); Tishby & Zaslavsky (2015) and references therein. The roots of this perspective trace back to the original works of Geiger et al. (1997); Tishby et al. (1999) which proposed that, ideally, the network extracts approximate sufficient statistics of the input with respect to the output, i.e., the network finds a maximally compressed mapping of the input variable that preserves as much of the information on the output variable as possible.

We also point out studies of complexity in a layer, not focusing on the concept of entropy, such as Montufar et al. (2014).

We note that in these previous studies, layers have often been treated as undifferentiated blocks. This perspective overlooks an important limitation: any information measure that remains unchanged under one-to-one rearrangements of these blocks fails to capture the internal organization of a layer. In particular, it ignores how the nodes are structured into feature dimensions or channels. This topic will be an aspect central to our approach.

Another topic addressed in this paper is the confidence measure associated with the output of a classification machine. A confidence measure quantifies how likely a prediction is to be correct. Although the standard softmax output score is often used, it can be misleading and overly confident. Here, we conduct a preliminary study to use the information content that we propose as a possible candidate for the confidence measure.

### 1.2 OUR WORK: GRANDMOTHER-ENTROPY

The starting hypothesis we make is that the activity of a cell in any layer is informative about the class associated with an input. This hypothesis is inspired by the idea of a "grandmother cell" in neuroscience Barlow (1972); Gross (2002) as well as pioneering work on the sparsity of cell activation values Field (1994); Olshausen & Field (1996). It has the intuition proposed early by Geiger et al. (1997); Tishby et al. (1999) that NNs compress the input to provide selected information to the classification output. Our approach considers the association of the activation value, given an input, with the probability that this node is informative to the class of input. In the extreme case, when a "Jennifer Aniston neuron" Connor (2005) is activated, classification information is immediately inferred. Also, clearly, the last layer of a classification NN is presented in this way, each class being associated with each cell activation. In the best-case scenario, only the correct node is activated in the output layer. In neuroscience the invariance of neuron responses to varying 3D object appearances has been observed Vetter et al. (1995).

Our approach associates with the larger activation of a given node, given an input, a larger probability that such a node is informative towards a classification of such input. Next, we develop an entropy for each layer of a network, associated with an input, that captures the information given within this structure, and we refer to it as the grandmother-entropy, or GM-entropy for simplicity. We then analyze the evolution of this entropy through the layers of the network.

As an application of this entropy we examine whether the entropy of the last convolution layers in a classification network can serve as a more reliable confidence measure than the output softmax values of the top classification unit. We emphasize that this is a preliminary study restricted to data corruption techniques applied to images and we obtain encouraging results.

.

### 1.3 ORGANIZATION

The paper is organized as follows. Section 2 presents our main definition of information, a grandmother-entropy. Section 3 performs an empirical study of the propagation of the grandmother-entropy through the layers of ResNet-50. Section 4 considers the grandmother-entropy of the last convolution layer of ResNet-50 as a candidate measure of confidence in the classification output. We emphasize that this is a preliminary study. Section 5 concludes the paper.

## 2 GRANDMOTHER-ENTROPY IN A NEURAL NETWORK

We are given a Neural Network (NN) layer, indexed by $l = 1, \ldots, L$ and with each feature, or channel, indexed by $d = 1, \ldots, D_l$ ($d$ for feature dimension). For each feature, the nodes are indexed by $j = 1, \ldots, N_l$, i.e., for all features of a layer $l$, there are $N_l$ nodes. For an image convolution layers are described by $N_l = w_l \times h_l$, where $w_l$ is the width and $h_l$ is the height of the feature layer. Thus, a node or vertex in an NN is indexed by the triplet $v = (j, d, l)$. Note that the total number of nodes in a layer $l$ is then the product $D_l \times N_l$. Given an input $x$ propagating through an NN, each node of the NN is activated with values $a_v = a(j, d, l; x)$.

A layer space in an NN is indexed by $(j, d)$. Note that at times the pair $(j, d)$ is to be interpreted as indices and at times as variables. Just as in physics, a position of a particle in a discrete lat-

tice can have these two interpretations. Throughout the paper, it should be self-explanatory which interpretation is the most appropriate.

We investigate the information content of the activation values $a_v$ in a layer, given an input $x$. We first must clarify what we mean by information. Our approach is to associate an activation of a node with a probability of a node. Here we stress that the activation is not to be interpreted as the outcome of a probability distribution but rather associated with the probability that a node, and not another node, is an outcome of a probability distribution over all nodes. Our approach is based on the idea that a node's activation value reflects the probability that the node is informative for the classification task associated with the input.

In our empirical studies, we consider the activation values as nonnegative quantities (such is the case with state-of-the-art ReLu activation functions, and it was also the case for early models using sigmoid functions). We associate the activation of a node $a(j, d, l; x) \in \mathbb{R}_+$ in a layer $l$ with a probability as follows.

$$a(j, d, l; x) \geq 0 \qquad \Rightarrow \qquad P(j, d; l, x) = \frac{1}{Z_l} a(j, d, l; x) . \tag{1}$$

where

$$Z_l = \sum_{d=1}^{D_l} \sum_{j=1}^{N_l} a(j, d, l) , \tag{2}$$

is a normalization constant that transforms all node activity into a probability distribution $P(j, d; l, x) \geq 0$ of a layer probability space $l$. Interpreting $(j, d)$ as random variables in a layer probability space indexed by $l$ and given an input $x$ implies a random process that selects integer values $(j, d)$ with probability $P(j, d; l, x)$. Note that for negative activation values, we would first map them to their magnitude or square values, so they become non-negative, but we have not performed an empirical study for this case.

We could now define the layer information as the Shannon entropy associated with the joint distribution $P(j, d; l, x)$. This entropy would be invariant to any one-to-one transformation of the layer variables $(j, d)$ to another same size set of two integer variables, as the probabilities in this new set of integer variables would not change. However, we argue that the information content of an NN needs to be based on the feature (or channel) architecture. Given an input $x$ to a network, it is informative that a feature, in a given layer $l$, responds well or not to a node in position $j$. Thus, we look for a layer information definition that reflects the feature structure and thus, it is not invariant to changes in the NN architecture.

We then proceed to factorize the layer probability space and, for simplicity and clarity, drop the input $x$ and the layer $l$ temporarily from the probability notation. We compute the marginal probability of each feature dimension of a layer

$$P(d) = \sum_{j=1}^{N_l} P(j, d) = \frac{1}{Z_l} \sum_{j=1}^{N_l} a(j, d, l; x) . \tag{3}$$

From this marginal distribution we obtain the conditional probability distribution for each node within a layer as follows.

$$P(j|d) = \frac{P(j, d)}{P(d)} = \frac{a(j, d, l; x)}{\sum_{j=1}^{N_l} a(j, d, l; x)} . \tag{4}$$

Thus, by Bayes theorem, the layer probability space is factorized into a feature probability space, with $P(d)-$ prior probabilities, and a configuration probability space, with $P(j|d)-$ conditional probabilities given a feature dimension (see figure 1 for a visualization of the factorization of the layer probability space.)

This factorization exposes the information structure of the layer; General one-to-one correspondence transformations of the set $\{j, d\}$ that mix nodes from different feature dimensions would cause different factorization probabilities of the joint distribution. We are now equipped to define the information content of a layer associated with this particular layer structure, given an input $x$, as the

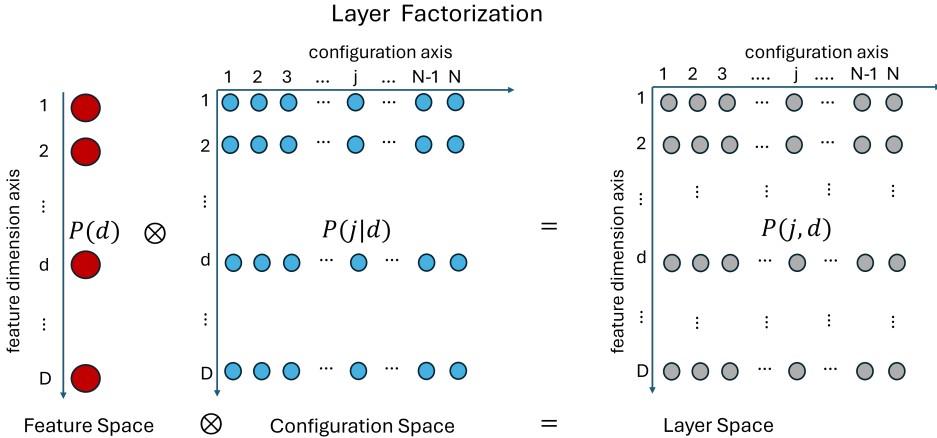

Figure 1: A visualization of the probability spaces associated with a layer of the NN. The layer space captures the joinXt?X probability $P(j,d)$ and can be factorized as a product of the feature probability space with the configuration probability space per feature dimension. This product of probability spaces requires the operation of one unit probability of the feature space to be multiplied by each unit probability of the configuration space per feature. While the entropy of $P(j,d)$ would be invariant under all one-to-one correspondence transformations of the variables $j, d$ into other two integer variable, the conditional entropy would not. The conditional entropy respects the particular factorization shown above. Note that the conditional entropy is invariant under a global scale of the layer activations.

conditional entropy

$$S_l^{\text{GM}}(x) = -\sum_d^{D_l} \sum_{j=1}^{N_l} P(j,d;l,x) \log_2 P(j|d;l,x)$$

$$= \sum_{d=1}^{D_l} P(d;l,x)\, S_{l,d}^{\text{configuration}}(x) \tag{5}$$

where the configuration entropy, per feature dimension, is given by

$$S_{l,d}^{\text{configuration}}(x) = -\sum_{j=1}^{N_l} P(j|d;l,x) \log_2 P(j|d;l,x)\,, \tag{6}$$

and we return the inclusion of the variables $l, x$ to the probabilities notation. The label GM stands for grandmother-entropy. Conditional probabilities $P(j|d;l,x)$ assign the likelihood that a node $j$, a layer $l$ and feature $d$, being activated from an input $x$, and not another node among the ones belonging to the feature layer $d, l$. Thus, conditional entropy assigns the expected information content to these likelihoods on all features of the layer $l$. Note that the information of a network layer is invariant under a global scale transformation of the activation, since the scale will reset the value $Z$ in equation 2 while the probabilities remain invariant.

Also note that it is common for NN's to have some special layers reduced in the configuration dimension while increasing in the feature dimension, as one moves forward to the output. When such changes occur, the conditional entropy has a bias towards reduction. For example, it is straightforward to verify that a uniform distribution across the entire layer will have the conditional entropy to be reduced with the configuration sizes reduction, regardless of if the feature dimension increases. This effect at these special layers deserves attention when we analyze the GM-entropy.

Figure 2 illustrates an example that compares, across the layers of a neural network, the behavior of the entropy of the joint distribution $P(j,d;l,x)$ with the conditional entropy $S_l^{\text{GM}}(x)$. The results support the empirical usefulness of GM-entropy as a measure of the information content of a network layer.

Next, we conduct an empirical study of how the information content $S_l^{\mathrm{GM}}(x)$ evolves as input $x$ propagates through the ResNet-50 layers. We also analyze the effect of residual connections on the information content at each layer. In this paper, our experiments focus on networks with pretrained weights and biases. Extending the study to examine how information evolves during the learning process is an important direction for future research.

# 3 AN EMPIRICAL STUDY OF THE EVOLUTION OF GM-ENTROPY THROUGH RESNET-50

Consider a trained classification network, specifically ResNet-50, which exemplifies state-of-the-art performance. ResNet is a convolutional neural network (CNN) that embodies a foundational layer technique that is now ubiquitous in deep learning: skip connections.

The input image $x$, is a color image comprising three channels (R, G, B). This image is processed through successive convolution layers in the ResNet architecture, incorporating ReLU activations, skip connections, and batch normalization, until an average pooling across the spatial dimension flattens the information and a fully connected layer produces the output where a softmax outputs a probability distribution $P(c|x)$, over the $C$ output nodes/classes.

Note that there are four layers where the spatial size is reduced and the feature dimension is increased, namely the layers 12, 14, 25, 27, 44, 46. This reduction creates a bias effect to reduce the GM-entropy.

In the first empirical study, we investigate the GM-entropy through each of the convolution layers after skip connections for ResNet-50 (see Figure 2) a. We observe that the GM-entropy decreases through most of the layers. This illustrative result is also verified by an experiment with the ImageNet21K (Winter 2021 Release) data set Deng et al. (2009); Deng et al., where we randomly sample 10 classes and 100 images per class, that is, 1000 images test. Table 1 indicates that the GM-entropy is decreasing.

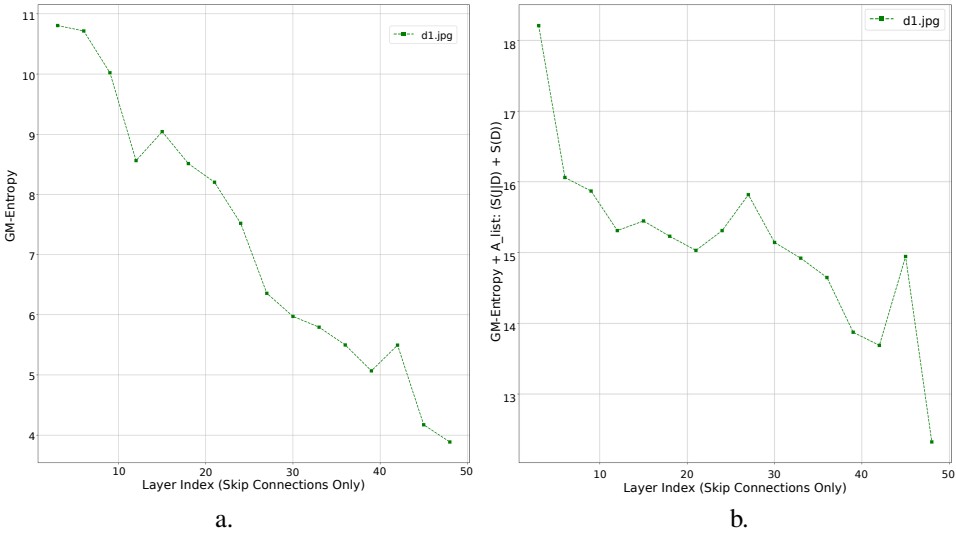

a.          b.

Figure 2: a.GM-entropy evolution through the convolution layers, for an input image shown in figure 7a d1.jpg. The GM-entropy is shown for all convolution layers, after skip connections and batch normalization. The GM-entropy decreases for all layers evolution with the exception of two layers. b. The entropy of the joint distribution $P(j, d; l, x)$. Note that while the GM-entropy decreases consistently through layers the joint distribution does not. The fact that GM-entropy is smaller than the joint entropy is a mathematical property, as their difference is the entropy of the marginal distribution $P(d; l, x)$.

We test the hypothesis that GM-entropy decreases with depth. For each image, we fit an OLS (ordinary least squares) regression of GM-Entropy vs. layer index (0–48) and test $H_0 : \beta_1 = 0$ vs.

$H_1 : \beta_1 < 0$. A one-sample $t$-test on slopes yields $t(999) = -1395.6$, $p < 10^{-300}$. A regression on the layer-wise mean curve (49 points) also shows a strong negative trend (slope $-0.1617$, $r = -0.976$, $p = 1.27 \times 10^{-32}$), providing strong evidence that GM-entropy decreases with depth in ResNet-50.

Table 1: Per-image OLS fit of GM-entropy vs. layer index (0–48) over $N = 1000$ images.

|  | Mean slope | Intercept | SD | 95% CI (slope) |
| --- | --- | --- | --- | --- |
| Per-image best-fit line | $-0.161708$ | $11.449416$ | $0.00366$ | $-0.161708 \pm 0.00023$ |

## 3.1 SKIP CONNECTIONS

In skip connection, the input is directly passed through to the next layer along with the output of the intermediate convolution layers. We hypothesize that, during the intermediate convolution layers, less information needs to be learned since the aggregation of the input will restore the information of the previous layer. To test this hypothesis, we investigate the GM-entropy during these intermediate layers and after the addition of the input, when the information is restored. The results seen in Figure 3 suggest that this hypothesis is empirically verified in ResNet-50. It suggests that between skip connections less information is learned, possibly providing an explanation for the efficiency of skip connection architecture.

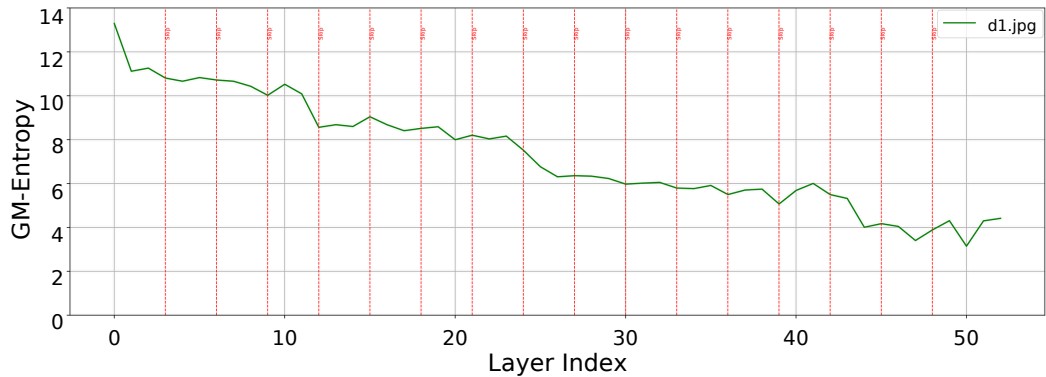

Figure 3: GM-entropy of activations after BatchNorm for each of the 53 layers in ResNet-50. The input image is the one shown in figure 7a d1.jpg. Vertical red dashed lines indicate skip-connection aggregation layers. It is noticeable that between the skip connections the GM-entropy many times increases to only decrease after the addition of the input. The vertical lines at layers 12, 14, 25, 27, 44, 46 indicate when a reduction in spatial size occurs.

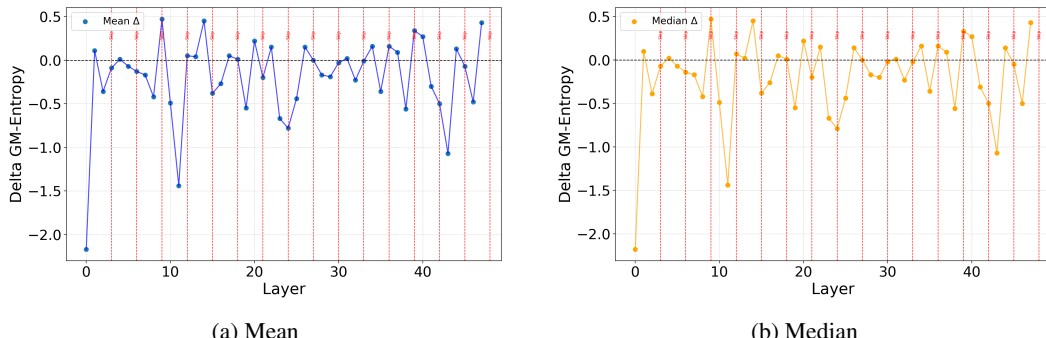

(a) Mean

(b) Median

Figure 4: The change of GM-entropy between layers in ResNet-50 tested on 1000 images from ImageNet21K, selected randomly from 10 classes and 100 images per class. a. The y-axis is the mean of the changes (Mean $\Delta$) over 1000 images vs the x-axis are each layer. b. The y-axis is the median of the changes (Median $\Delta$) over 1000 images vs the x-axis are each layer. Clearly the changes alternate with positive changes between skip connections.

## 4  TOWARDS A CONFIDENCE MEASURE

We start by examining to what extent the GM-entropy of the last convolutional layer of ResNet-50 varies as the input is corrupted by blur or noise. Note that the last convolutional layer is the layer before the penultimate layer in the sense that after that there is a global average pooling followed by a fully connected layer to the output layer.

So, the first set of experiments we did was to consider effects on the GM-entropy by blurring an image (see figure 5 a.) or by adding noise to an image (see figure 5 b.) and examining the GM-entropy evolution (see figure 6). For all layers, the GM-entropy increases as the blur increases or as the

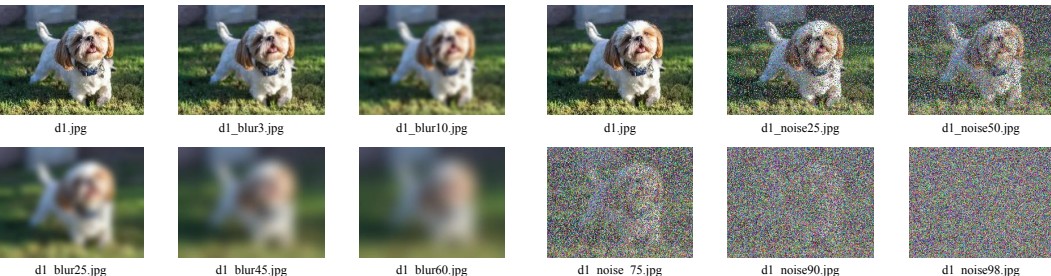

Figure 5: A picture of a Shih Tzu dog, d1.jpg, corrupted by various degrees of a. blur or b. noise.

noise increases. The last convolution layer, layer 48, is where the effect is most accentuated. Thus, we now explore further the hypothesis that the GM-entropy of layer 48 can provide a confidence measure focusing on the GM-entropy values at the last convolution layer. The next experiment we consider five pictures of dogs from ImageNet21K. We apply blur at various levels to all pictures and evaluate GM-entropy of layer 48 as a candidate for a confidence measure. We then compute the softmax value of the classification unit, often considered a confidence measure of the classification, and create a graph of these two confidence measures (see figure 7 a.). Similarly we apply noise at various levels and produce the same graph (see figure 7 b.) This comparison suggests that for these corruptions of images the GM-entropy outperforms the softmax measure.

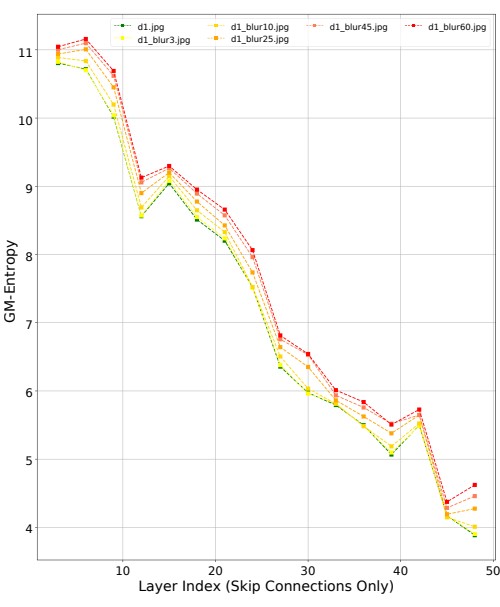

(a) Blurred images. Color encodes blur level on the python code applied-blur-to-image with blur levels (0, , 10, 25, 45, 60).

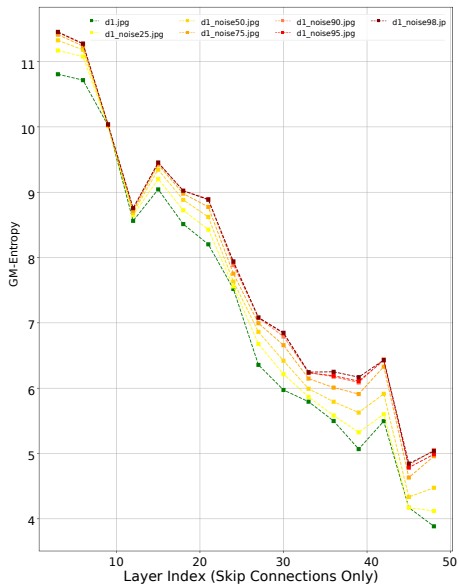

(b) Noisy images. Color encodes noise level (0%, 25%, 50%, 75%, 90%, 95%, 98%), where this numbers are the percentage of pixels replaced by random values.

Figure 6: GM-Entropy at skip-connection after BatchNorm layers. Clearly, the GM-entropy increases at all layers the more the images are corrupted either by blur or noise.

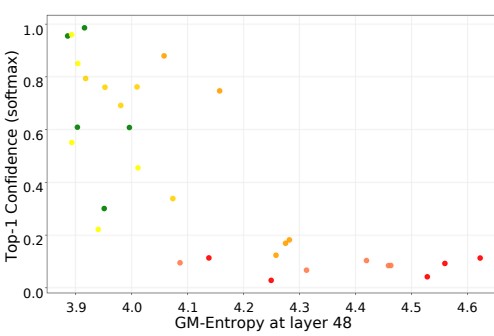

(a) Color coded blurred images displayed in a graph of confidence of Top-1 Classification vs GM-Entropy

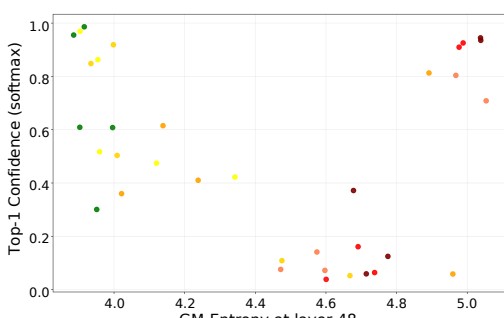

(b) Color coded noisy images displayed in a graph of confidence of Top-1 Classification vs GM-Entropy

Figure 7: We considered five pictures of dogs from ImageNet21K, so each color in the graph above is shown five times. We applied to these five pictures the same levels and same color coded as figure 6 in a. blur at five levels and in b. noise at six levels. We then evaluate GM-entropy of layer 48 as a candidate for a confidence measure. For all these data, a. and b. we then compute the softmax value of the top-1 classification unit, often considered a confidence measure of the classification, and create a graph with axis of these two confidence measure (see figure 7 a. and b. ). Clearly, the higher is the corruption of the image the higher is the GM-entropy while the top-1 softmax produces high values for large amount of corruption (and wrong classification results). The GM-entropy outperforms the softmax measure for these corruptions.

Table 2: We study the impact on the GM-entropy of blurring or adding noise to an image over 1000 images from ImageNet21k (10 random classes and 100 images per class). In this case, we fixed at the third level of the previous experiments, namely the blur of python code applied-blur-to-image at blur level 25 and the noise level at 75%. We focused on layer 48 (the last convolution layer) and evaluated the GM-entropy. We compare with the GM-entropy of the original image.

| Image Corruption Method | mean | median | std |
|---|---|---|---|
| original image | 4.024042 | 4.017071 | 0.182951 |
| blur level 25 | 4.221479 | 4.216569 | 0.137546 |
| noise added 75% | 4.614819 | 4.615324 | 0.199378 |

## 5  CONCLUSION

We introduced GM-entropy as a measure of information associated with each layer of a deep neural network for a given input. GM-entropy interprets each activation unit as contributing to an information probability: higher activations correspond to a greater likelihood of conveying information. Unlike the entropy of the joint distribution over a layer, GM-entropy is defined as a conditional entropy over the feature dimension, and thus it is sensitive to transformations that mix feature units rather than invariant to them.

On the empirical side, we evaluated the GM-entropy on ResNet-50, a widely used benchmark in large-scale image classification, and showed that it decreases consistently between layers. It was also observed that during the skip connection, before the aggregation of the input, the GM-entropy increased to later decrease with the input aggregation. This suggests that a possible explanation for the efficiency of skip connections is the management of the information. We conducted preliminary experiments on how GM-entropy responds to image corruption, such as increasing levels of blur or noise. Our results indicate that the last convolutional layer (layer 48) is particularly informative: GM-entropy rises systematically with the severity of corruption, effectively discriminating between degrees of degradation. In contrast, softmax confidence values can remain high even under heavy corruption, often leading to incorrect classifications.

In future work, we think that more studies of the GM-entropy for all other AI domains would be interesting, as well as a more thorough study of the GM-entropy values at the last layer of convolution may lead to better confidence measures for classification.

### AUTHOR CONTRIBUTIONS

Equal contribution of the authors.

### REPRODUCIBILITY

We plan to make the code available in GitHub and the dataset is already publicly available.

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
