# OpenReview forum: "On the Decrease of Grandmother-Entropy in AI"
_ICLR.cc/2026/Conference — ICLR 2026 Conference Withdrawn Submission_

### Official Review · Reviewer_cN7N · 2025-10-16

**Soundness:** 1
**Presentation:** 2
**Contribution:** 1
**Rating:** 0
**Confidence:** 4

**Summary:**

The paper proposes "grandmother entropy", an information-theoretic quantity that assesses the importance of individual neurons for the classification of single data points. The paper shows that grandmother entropy decreases throughout layers, and that it presents a reliable confidence measure for classification.

**Strengths:**

The paper treats an interesting and timely topic, as information-theoretic analyses of neural networks are a promising avenue to understanding deep learning. The concept is novel to the best of my knowledge.

**Weaknesses:**

Unfortunately, the paper has several shortcomings that prevent me from recommending acceptance at this stage:
- The concept of grandmother entropy seems ad hoc, and the choice to simply normalize activations linearly is arbitrary. Indeed, the same normalization to a sum of 1 can be achieved by passing all activations through a softmax layer. Why was the linear normalization chosen, and not any other one? This must be explained.
- The paper misses links to the literature that study the importance of individual neurons for classification tasks. Indeed, there have been measures proposed for class selectivity such as (1803.06959) or (1804.06679). The present work must be put in the context of existing results.
- The experimental evidence is, at the moment, quite weak. While the paper presents several lines of investigation---change with layer number, effect of residual connections, sensitivity to noise and blurring, correlation to confidence---these results are all obtained on a single network architecture, and sometimes even on just a single data sample (Fig. 2, 3). Thus, at best, the experimental evidence is only anecdotal. This is insufficient for a top-ranked conference, in my opinion.
- The claim that GM entropy is outperforming softmax as a confidence measure is not supported. Indeed, while Fig. 7b shows that softmax confidence can be high for noisy images, at the same time we see that GM entropy also does not distinguish noisy and noise-free images perfectly. Specifically, the orange dots in Fig. 7b span almost the full range of GM entropy, and so do the red dots in Fig. 7a. More convincing evidence must be presented for this claim.
- The writing style of the paper is subpar and must be improved before the paper can be accepted at a top-ranked conference.

**Questions:**

No questions at the moment.

---

### Official Review · Reviewer_cw3e · 2025-10-28

**Soundness:** 2
**Presentation:** 1
**Contribution:** 2
**Rating:** 2
**Confidence:** 5

**Summary:**

In the paper, the authors propose an innovative approach: conditional entropy is employed to quantify the inter-channel or inter-feature layer information, which they term "Grandmother Entropy". Empirical evaluations conducted on ResNet-50 demonstrate that as input images propagate forward through the network layers, the Grandmother Entropy exhibits a decreasing trend. Furthermore, the Grandmother Entropy of the final convolutional layer serves as a reliable confidence metric for the classification results.

**Strengths:**

The author's ideas are remarkably novel and highly innovative. In the paper, the author explores activation values from an alternative perspective and frames them as a form of confidence measure.

**Weaknesses:**

The current study suffers from a critical limitation in its experimental scope: the inclusion of only ResNet-50 and partial experiments on ImageNet is far from sufficient to substantiate its core claims. Additionally, the writing quality of the manuscript is subpar, lacking in clarity, rigor, and coherence. To enhance the credibility and academic value of this work, the authors are strongly advised to expand their experimental design—specifically, by testing a broader range of activation functions (e.g., ReLU variants, Swish, GELU) and incorporating more diverse network architectures . This expanded experimentation would not only provide more comprehensive evidence to validate the proposed approach but also enable meaningful comparisons with existing state-of-the-art methods, thereby strengthening the overall persuasiveness of the research.

**Questions:**

Could the author please explain why the magnitude of activation can represent probability? In the case of the leaky-ReLU function, negative activation values and positive activation values hold equal importance.

---

### Official Review · Reviewer_bT6Q · 2025-10-30

**Soundness:** 3
**Presentation:** 3
**Contribution:** 2
**Rating:** 4
**Confidence:** 4

**Summary:**

The authors propose a new way to compute how the entropy of activations changes in different parts of the deep neural network. They perform tests on a small subset of imagenet and one resnet model.

**Strengths:**

* The authors aimed to propose a method to compute entropy within neural networks that takes into consideration the internal organization of activations across channels and neurons.
* The paper is easy to follow.

**Weaknesses:**

* The idea to use entropy to compute information within the neural networks is not new. The majority of findings (e.g. that entropy decreases with layers) are not new and have been observed in earlier works (e.g. [1][2]) and are already widely accepted. While it can be a more focused approach, I do not see that it brings something completely new and different when it comes to the observations.
* The authors describe in detail things that are already a common knowledge in the area of deep learning, e.g. “By deep learning, we mean deep hierarchical neural networks trained with gradient descent.”, also “The input image x, is a color image comprising three channels (R, G, B). This image is processed through successive convolution layers in the ResNet architecture, incorporating ReLU activations, … output nodes/classes.”. They also focus very much on skip connections “A common technique within this framework — skip connections — have become standard in most state-of-theart models.” as it was the only important technique in the neural network domain, while it is already quite old.
* In our empirical studies, we consider the activation values as nonnegative quantities (such is the case with state-of-the-art ReLu activation functions - in the current networks this is not always the case - see all the GELU, ELU etc. activations (e.g. GELU is used in the ‘refreshed’ resnet - convnext).
* The experiments are very limited - they only test one CNN (ResNet) on 10 classes (with 100 images) from ImageNet. It would be nice to at least have more CNNs, preferably also ViTs - as two leading architectures. Moreover, the authors do not say what 10 classes they use. It would be better to perform multiple experiments with different sets of classes and an experiment with more examples and classes (e.g. using the mini-Imagenet dataset with 100 Imagenet classes).

**Questions:**

* The majority of findings (e.g. that entropy decreases with layers) are not new and have been observed in earlier works. Could the authors highlight what new can be discovered with their method and cannot be discovered with other entropy measures?
* Please refer to the weaknesses.

---

### Official Review · Reviewer_ADMo · 2025-10-30

**Soundness:** 2
**Presentation:** 2
**Contribution:** 1
**Rating:** 2
**Confidence:** 4

**Summary:**

The paper introduces Grandmother-Entropy (GM-entropy), a conditional entropy-based measure designed to quantify the information content within neural network layers. The authors show empirically that GM-entropy decreases across the layers of ResNet-50, and they propose its potential use as a confidence measure in classification, comparing it to softmax-based confidence.

**Strengths:**

The concept is novel in that the proposed GM-entropy is an interesting and original idea linking neuroscience-inspired “grandmother cell” intuition with information-theoretic analysis of deep networks.

The observation that GM-entropy consistently decreases through layers provides a potentially meaningful interpretability metric.

The manuscript is generally well-written and easy to follow, with theoretical explanation followed by empirical evaluation.

**Weaknesses:**

The connection to neuroscience concepts such as “grandmother cells” is largely metaphorical and not mathematically grounded.

Experiments are limited to ResNet-50 and only consider ImageNet images.

The authors do not test the generality of the entropy trend across architectures (e.g., transformers, MLPs, Mambas) or on different datasets.

The claim that GM-entropy “outperforms softmax” is based on qualitative plots without quantitative metrics.

No comparison is made to other uncertainty estimation methods.

It remains unclear whether GM-entropy reflects information compression, sparsity, or feature selectivity.

The relationship between decreasing GM-entropy and representation quality is not established. Does lower entropy imply better generalization or overconfidence?

**Questions:**

How does GM-entropy behave during training (not only inference)?

Would GM-entropy increase again in overfitted networks or under adversarial perturbations?

Can GM-entropy be extended or normalized to allow comparison across architectures of different depths?

How sensitive is the entropy measure to activation scaling or normalization layers?

**Details Of Ethics Concerns:**

No ethics concerns.

---

### Note · Authors · 2025-11-13

I have read and agree with the venue's withdrawal policy on behalf of myself and my co-authors.